# Difference Analysis Among Six Kinds of Acceptor Splicing Sequences by the Dispersion Features of 6-mer Subsets in Human Genes

**DOI:** 10.3390/biology14020206

**Published:** 2025-02-15

**Authors:** Yangming Si, Hong Li, Xiaolong Li

**Affiliations:** Inner Mongolia Autonomous Region Key Laboratory of Biophysics and Bioinformatics, School of Physical Science and Technology, Inner Mongolia University, Hohhot 010021, China; siyangming1991@163.com (Y.S.); ndlixiaolong@126.com (X.L.)

**Keywords:** human gene, acceptor splicing sequence, classification of splicing modes, XY1 6-mer subset, dispersion feature, difference analysis

## Abstract

We first constructed a comprehensive dataset of human gene acceptor splicing sequences, which were classified into common splicing, constitutive splicing, and alternative splicing modes. Alternative splicing was further subdivided into normal, exonic, and intronic sub-modes. We used 16 dispersion features of XY1 6-mer subsets to uncover the differences in the sequence composition among the six kinds of splicing modes. We found that the dispersion features can effectively distinguish the differences among the three modes and among the three sub-modes. Our study indicates that the dispersion features of 6-mer subsets can reveal the differences in base correlation in acceptor splicing sequences. In exploring base correlation, our method is important for studying functional sequences.

## 1. Introduction

In eukaryotes, the majority of protein-coding genes are organized into alternative introns and exons. During transcription, introns are removed from the precursor mRNA, a crucial process that ensures the mature mRNA can be translated into a polypeptide chain, ultimately forming a functional protein [1]. Recently, in vivo experimental studies have demonstrated that the catalytic spliceosome enzymes are spatially and temporally associated with RNA polymerase II, suggesting that spliceosome assembly and catalytic activity are closely coordinated with RNA transcription [2]. These observations support a potential synergy between transcription and splicing, thereby establishing a co-transcriptional splicing mechanism.

Splicing events occur at two specific positions within the sequence: the upstream position, known as the donor splice site; and the downstream position, referred to as the acceptor splice site. Based on the specific regions where these splice sites are located, splicing events can be classified into constitutive splicing and alternative splicing. The pairing of donor and acceptor sites from these two types can result in various splicing modes [3]. Constitutive splicing occurs exclusively at exon–intron boundaries, whereas alternative splicing can occur not only at exon–intron junctions but also within intronic or exonic sequences [4,5,6].

Previous studies indicate that the canonical donor–acceptor splice pattern in eukaryotes is GT-AG. In mammals, approximately 99.24% of donor–acceptor splice sites follow the canonical splicing pattern [7,8,9], while in plants, this proportion is approximately 98.7% [10]. Beyond the GT-AG pattern, other non-canonical splicing modes also exist [11,12,13,14,15,16,17,18,19,20,21,22,23,24,25,26,27,28,29,30,31]. Canonical splicing relies on the dinucleotide sequences GT at the donor (+1 and +2 sites) and AG at the acceptor (−2 and −1 sites) splice sites, which are essential for the precise interaction with the U2 spliceosome. The integrity of these donor and acceptor sites is crucial for proper splicing, and disruptions, such as canonical splice site variants, can impair precursor mRNA-spliceosome interactions. Such impairments often lead to aberrant splicing events, potentially resulting in alterations or loss of gene function [32,33,34,35,36].

Predicting alternative splice sites is a key focus of current research. The maximum entropy principle (MEP) is an effective approach for detecting the strength of 3′ (acceptor) splice sites and splicing signals [37]. SpliceAI, a 32-layer deep neural network, predicts splicing events from precursor RNA sequences [38]. AGAIN identifies specific mutation sites between the splice branch point and the canonical acceptor splice site that may disrupt normal gene splicing [39]. The COSSMO model combines convolutional layers, long short-term memory (LSTM) networks, and residual networks to predict the Percent Spliced In (PSI) of alternative acceptor splice sites [40].

Our previous research analyzed the sequence composition of human acceptor splicing regions using the separability feature of XY1 6-mer subsets. The 6-mer feature can provide more comprehensive information than mononucleotide and dinucleotide analysis. We found that the separability features can significantly distinguish the sequence differences among alternative, constitutive, and common splicing modes, and among alternative splicing sub-modes (normal, exonic, and intronic). The results indicate that different splicing modes exhibit significant sequence preferences in the upstream and core regions of the acceptor splicing sequences, closely related to the recognition of splicing signals. It is known that functional sequences, such as splicing sequences, exhibit long-range base correlations and 6-mer features include information on long-range base correlations.

Although the separability features can effectively detect the base bias in upstream and core regions of the acceptor splicing sequences, they may not clearly detect the base bias in downstream regions of splicing sequences. We know that the sequence composition is reflected by both base bias information and base correlation information. In the downstream region of the acceptor splicing site, we consider that the base correlation information is dominant. In this study, we used the dispersions of XY1 6-mer subsets to characterize the base correlation feature and to explore the sequence composition and sequence differences in the acceptor splicing region, thereby offering new insights and methodologies for the identification and prediction of diverse splicing sequences.

## 2. Materials and Methods

### 2.1. Classification of Acceptor Splicing Sequence

Based on canonical splicing events (GT-AG pattern), we classified acceptor splicing sequences into three main modes: common acceptor splicing, alternative acceptor splicing, and constitutive acceptor splicing. The definition of the three main modes is shown in Figure 1.

Constitutive splicing occurs exclusively at exon–intron boundaries, whereas alternative splicing can occur not only at exon–intron junctions but also within intronic or exonic sequences across multiple gene transcripts [4,5,6]. In a splicing event involving both donor and acceptor sites, if both splicing sites belong to constitutive splicing, the splicing event at the acceptor site is defined as the common acceptor splicing mode and the donor site is defined as the common donor splicing mode (Figure 1A). If the acceptor site is constitutive splicing while the donor site is alternative splicing, the acceptor site is defined as the constitutive acceptor splicing mode (Figure 1C), and the donor site is defined as the alternative donor splicing mode. If the acceptor site is alternative splicing while the donor site is constitutive splicing, the acceptor site is defined as the alternative acceptor splicing mode (Figure 1B), and the donor site is defined as the constitutive donor splicing mode. At the acceptor splicing site, the common mode is a one-to-one splicing event, the constitutive mode is a one-to-more splicing event, and the alternative mode is a more-to-one splicing event for multiple gene transcripts.

The alternative acceptor splicing mode is further classified into three sub-modes (Figure 1B). If the alternative acceptor site is located at the intron-exon junction, the splicing is defined as the normal splicing. If the alternative acceptor site is located within an exonic region, the splicing is defined as the exonic splicing. If the alternative acceptor site is located within an intronic region, the splicing is defined as the intronic splicing.

### 2.2. Datasets

The splicing site annotation information for human genes was retrieved from the GENCODE v40 database (https://www.gencodegenes.org/human/ [accessed on 24 April 2023]) [41] and the COSSMO dataset (http://cossmo.genes.toronto.edu/ [accessed on 10 March 2023]) [40]. Annotation data for common donor and acceptor splicing sites were extracted from the GENCODE v40 database, while alternative acceptor sites with their corresponding constitutive donor sites and constitutive acceptor sites with their corresponding alternative donor sites were identified using both the GENCODE v40 and COSSMO datasets. To define the splicing regions, we extracted 100 bp DNA sequences centered on each acceptor splicing site, with 50 bp upstream and 50 bp downstream of the site, based on the GRCh38 human genome reference. To ensure the accuracy of sequence analysis, splicing events were excluded if the distance between donor and acceptor splicing sites was less than 100 bp, or if multiple alternative sites corresponding to the same constitutive site were separated by less than 100 bp. Annotation data confirmed the classification of acceptor splicing into three main modes: common, alternative, and constitutive splicing. Alternative splicing, as expected, exhibited three sub-modes: normal (at exon-intron borders), exonic (within exons), and intronic (within introns). The datasets from the COSSMO database were expanded using GENCODE v40 annotations, ensuring comprehensive coverage. The numbers of selected acceptor splicing sequences for each category are summarized in Table 1.

### 2.3. 6-mer Classification

All splicing sequences were aligned and centered on the splicing site, with 50 bp sequences upstream and downstream extracted. Using a 6 bp sliding window with a 1 bp step size, all possible 6-mer fragments were extracted, starting from the −50 bp upstream base. This process yielded 95 sets of 6-mers corresponding to each site from −50 to +45 relative to the splicing site. For a splicing mode containing N sequences, this process results in an N × 95 matrix, where each row represents the 6-mer data of a single sequence, and each column corresponds to a specific position.

The total number of possible 6-mers is 4^6^ = 4096. Based on their sequence characteristics, these 4096 6-mers can be categorized into different subsets. If a 6-mer contains a specific dinucleotide (XY), it is classified as an XY1 6-mer. Conversely, if the 6-mer does not contain the XY dinucleotide, it is classified as an XY0 6-mer. In this classification system, X and Y represent any of the four bases (A, C, G, or T), and the total set of 6-mers is divided into 16 groups, each consisting of an XY1 6-mer subset and an XY0 6-mer subset. When X ≠ Y, there are 2911 6-mers in the XY0 subset and 1185 6-mers in the XY1 subset. When X = Y, the XY0 subset contains 3105 6-mers, while the XY1 subset contains 991 6-mers. Under this classification, 16 kinds of XY1 6-mer subsets and 16 kinds of XY0 6-mer subsets are obtained.

Finally, we calculated the frequency of each 6-mer across all sites. Additionally, we analyzed the frequency of 6-mers within the XY1 subsets to investigate the distribution and prevalence of specific sequence motifs at splicing sites. This frequency analysis offers further insights into the functional significance of 6-mers in RNA splicing.

### 2.4. Definition of Dispersion Value

For the total 6-mers and each XY1/XY0 6-mer subset, the average frequency and the standard deviation were calculated. The dispersion (*ρ*_XY_) value was defined to quantify the distribution discreteness of XY 6-mer frequencies. The definition is as follows:*ρ*_XY_ = *σ*_XY_/*σ*

where *σ*_XY_ represents the standard deviation of the frequency distribution in an XY 6-mer subset, while *σ* represents the standard deviation of the frequency distribution in the total 6-mers. If *ρ*_*X**Y*_ < 1, then the standard deviation of the frequency distribution in the XY 6-mer subset is lower than that in the total 6-mers. This indicates that the usage of the XY 6-mer is more conservative. If *ρ_XY_* > 1, then the usage of the XY 6-mer is more random.

### 2.5. Statistical Analysis

Based on the 6-mer set at each site from sites −50 to +45 in the acceptor splicing sequences, we obtained the corresponding dispersion values of the XY1 6-mer subset at each site. For a given region in the acceptor splicing sequences, we can compare the differences in dispersion values of this region between different splicing modes. Here, paired sample *t*-tests were performed to assess the significance of differences. If the selected site number is N in the region, the degree of freedom for the paired sample *t*-test is N-1. We applied the Benjamani–Hochberg false discovery rate (FDR) correction to adjust the *p*-value, ensuring the robustness of the results and minimizing the risk of Type I errors. We used *p* < 0.05 as the significance threshold for statistical tests.

## 3. Results

### 3.1. Sequence Analysis of Acceptor Splicing Region

#### 3.1.1. Dispersion Distribution of XY 6-mer Subsets

The base composition of DNA sequences is non-random, characterized by two primary aspects: the frequency of bases that indicates the usage bias of bases and the frequency of base segments that indicates the base correlation [42]. In functional sequences, the range of base correlations is longer, forming biologically functional signal units [43]. It is generally believed that the range of base correlations is between 3 and 8 bp [44]. Therefore, analyzing the composition patterns of k-base segments (k-mer) in DNA sequences is an effective method to identify functional signals. K-mer features capture both usage bias of bases and base correlations. For splicing sequences, short k-mers (k = 1.2) cannot provide sufficient information, especially the information of long-range base correlation. However, long k-mer analysis will increase the analytical complexity. For instance, with k = 6, there are 4^6^ = 4096 possible 6-mers, making it challenging to identify general composition patterns from so many frequencies. Based on these considerations, to reduce complexity, we classified all 6-mers into 16 XY1 6-mer subsets containing the XY dinucleotide and 16 XY0 6-mer subsets not containing the XY dinucleotide. The dispersion values of these 32 6-mer subsets were used to explore the compositional information of acceptor splicing sequences.

For canonical splicing sites (GT-AG), we extracted a total of 313,303 acceptor splicing sequences. We selected 50 bp in length both upstream and downstream of the splicing site as the acceptor splicing region. Here, bases A and G are located at sites −2 and −1, respectively. All splicing sequences were aligned centered on the splicing site. Starting from the base site −50, we extracted 6-mer fragments at each site using a 6 bp window with 1 bp step size and obtained 95 6-mer sets from sites −50 to +45. The number of 6-mers in each 6-mer set is 313,303. For each site, the dispersion values of 32 XY 6-mer subsets were calculated and marked at the first base of the 6-mer. Finally, the dispersion distributions across the acceptor splicing sequence are shown in Figure 2.

The dispersion trends of the XY1 subset are opposite to those of the XY0 subset, due to the non-random sampling of the total 6-mer (Figure 2). In subsequent analyses, we focus exclusively on the dispersion of the XY1 subset. Based on the trend analysis of dispersion distribution, we defined the region from sites −50 to −8 as the upstream region, which includes the sequence information from base sites −50 to −3. The region from sites −7 to +1 is defined as the core region, which includes the sequence information from base sites −7 to +6. The region from sites +2 to +45 is defined as the downstream region, which includes the sequence information from base sites +2 to +50.

According to the definition, the dispersion value of the XY1 6-mer subset at a site reflects the conservation level of XY1 6-mer usage. A lower dispersion value indicates a narrower frequency distribution containing the XY dinucleotide in the 6-mer subset, suggesting higher conservation or stronger base correlation at the site referring to the 6-base segment. Conversely, a higher dispersion value implies a weaker base correlation or more random usage of XY1 6-mers at the site.

We observe a distinct non-uniform distribution of dispersion values for the 16 XY1 6-mers within the acceptor splicing region (Figure 2). Each subset displays distinct dispersion preferences across the upstream, core, and downstream regions, indicating that dispersion characteristics could effectively reveal the compositional information in the acceptor splicing sequences.

#### 3.1.2. Upstream Region of Acceptor Splicing Sequences

There is an obvious variation in the dispersion distribution of XY1 6-mers within the upstream region, specifically from sites −30 to −8 (Figure 2). The dispersion values of 12 6-mer subsets (AA1, GG1, CC1, AG1, GA1, AT1, TA1, AC1, CA1, GT1, TG1, and GC1) are obviously lower than the background value, indicating strong conservation of these 6-mer subsets or stronger base correlation in the region (Figure 2A–F,I–L,M,O). The dispersion values of the CT1 and TC1 6-mer subsets are close to the background value, suggesting their base correlation is similar to the background distribution (Figure 2H,N). The dispersion values of the TT1 6-mer subset are obviously higher than the background value, indicating that the usage of TT1 6-mers is more random in this region (Figure 2P). The dispersion of the CG1 6-mer subset remains consistently low (~0.3) across the entire acceptor splicing region (Figure 2G), indicating that CG1 6-mers are not the major motifs related to splicing signals. The researchers indicated that it is associated with genomic evolution and the structure of CpG islands as well as the nucleosome sequences [45,46,47].

#### 3.1.3. Core Region of Acceptor Splicing Sequences

In the core region, the dispersion distribution of the 16 XY1 6-mers shows the occurrence of obvious variation and extreme dispersion values. At sites −6 and −5, the dispersion values of AA1, AG1, AT1, GA1, GG1, and GT1 6-mer subsets are obviously lower than the background value (Figure 2A,C,D,I,K,L), indicating that the base conservation or base correlation is strong in the core region from sites −6 to +1. This indicates that the core region is crucial for identifying the acceptor splicing site. As in the description of the upstream region, the strong base conservation of CG1 6-mers is not a splicing signal. For the other 9 XY1 6-mers, the base conservation is not obvious, though the dispersion values of these 6-mer subsets show some variations in the core region.

#### 3.1.4. Downstream Region of Acceptor Splicing Sequences

Our previous study showed that there are no obvious local base biases in the downstream region of the acceptor splicing site characterized by the separability feature. When the sequence composition is characterized by the dispersion feature in this region, the dispersion distributions show obvious variations (Figure 2). The dispersion values of AA1, AC1, AT1, and TT1 6-mer subsets are obviously lower than the background value in some local regions, indicating strong conservation of these 6-mer subsets or stronger base correlation in these local regions (Figure 2A,B,D,P). The dispersion values of AG1, CT1, GC1, GG1, and TG1 6-mer subsets are obviously higher than the background value generally in the downstream region, suggesting a more random composition of these 6-mers in this region (Figure 2C,H,J,K,O). The dispersion values of CA1, CC1, GA1, GT1, and TC1 6-mer subsets fluctuate around the background value in the downstream region (Figure 2E,F,I,L,N). The dispersion values of CG1 and TA1 6-mer subsets remain consistently lower than the background value across the entire downstream region, without obvious changes (Figure 2G,M). The dispersion distributions of the two 6-mer subsets reflect the general sequence composition feature or general base correlation feature of the human genome and the feature is related to genome evolution [45], indicating that CG1 and TA1 6-mer subsets are not involved in the acceptor splicing signal.

### 3.2. Differences Analysis of Three Acceptor Splicing Modes

The acceptor splicing sequence set was classified into common mode, constitutive mode, and alternative mode. The corresponding sequence numbers are 57,938, 48,414, and 206,951, respectively. Then, we obtained the dispersion distributions of 16 XY1 6-mers in three acceptor splicing modes. The dispersion distributions are shown in Figure 3.

The dispersion distributions of XY1 6-mer subsets (Figure 3) for the common and constitutive splicing modes are similar to the distributions in the total samples (Figure 2). However, the dispersion distributions in the alternative splicing mode are obviously different from the other two modes. Generally, the dispersion distributions of most XY1 6-mer subsets showed no obvious differences in the core region among the three splicing modes, indicating the sequence composition exhibits a highly conserved pattern. Compared with the common and constitutive modes, the dispersion values of the CT1, GT1, and TT1 6-mer subsets showed obvious differences in the alternative mode (Figure 3H,L,P). This indicates that, despite the sequence composition in the core region being highly conserved, certain 6-mer subsets demonstrate the sequence-specific variability among the three splicing modes. In contrast, obvious differences in the dispersion distributions appeared in the upstream and downstream regions among the three splicing modes. To explore the sequence composition differences among the three splicing modes, we performed a paired sample *t*-test on the dispersion values of XY1 6-mer subsets at each site within the upstream, core, and downstream regions.

#### 3.2.1. Upstream Region of Three Acceptor Splicing Modes

We extracted the dispersion values of XY1 6-mer subsets in 44 sites from sites −50 to −7 to perform the paired sample *t*-test among the three kinds of splicing modes. The average dispersion values of XY1 6-mer subsets in the upstream region and the results of the significant test are shown in Figure 4.

In the upstream region, the number of the compared dispersion values is 44; thus, the degree of freedom for the paired sample *t*-tests is 43. The paired sample *t*-tests showed that the dispersion differences in eleven subsets are significant among the three acceptor splicing modes. The eleven subsets are AC1, AT1, CA1, TA1, CG1, GC1, GT1, TG1, CT1, CC1, and TT1 6-mers (Figure 4A). For the AA1, GG1, and TC1 6-mer subsets, the paired sample *t*-tests showed that the dispersion differences significantly distinguish the alternative mode from the common/constitutive modes (Figure 4B). The three kinds of splicing modes cannot be distinguished significantly by the dispersion values of AG1 and GA1 6-mer subsets (Figure 4C). These results suggest that the sequence composition in the upstream region differs among the three splicing modes.

#### 3.2.2. Core Region of Three Acceptor Splicing Modes

In the core region, we extracted the dispersion values of XY1 6-mer subsets in 7 sites from sites −6 to +1 to perform the paired sample *t*-test. The degree of freedom for the paired sample *t*-test is 6. The results are shown in Figure 5.

For the CA1 and TC1 6-mer subsets, the paired sample *t*-tests showed that the dispersion differences are significant among the three acceptor splicing modes (Figure 5A). For the AC1 and CT1 6-mer subsets, the paired sample *t*-tests indicated that the dispersion differences only significantly distinguish the constitutive mode from the other two. For the GA1 and TT1 6-mer subsets, the paired sample *t*-tests indicated that the dispersion differences only significantly distinguish the alternative mode from the other two (Figure 5B). For the other ten XY1 6-mer subsets, the dispersion values cannot significantly distinguish the differences among the three acceptor splicing modes (Figure 5C). The results indicate that the sequence composition in the core region is more conserved, but there are still some differences among the three splicing modes.

#### 3.2.3. Downstream Region of Three Acceptor Splicing Modes

In the downstream region, we extracted the dispersion values of XY1 6-mer subsets in 44 sites from sites +2 to +45 to perform the paired sample *t*-test. The degree of freedom for the paired sample *t*-test is 43. The results are shown in Figure 6.

In the downstream region, the paired sample *t*-tests showed that the dispersion differences in nine XY1 6-mer subsets are significant among the three acceptor splicing modes. The nine subsets are AA1, AT1, CA1, CT1, GC1, GT1, TC1, TG1, and CG1 6-mers (Figure 6A). For AG1, GA1 and TA1 6-mer subsets, the paired sample *t*-tests showed that the dispersion differences only significantly distinguish the alternative mode from the other two modes (Figure 6B). For AC1, CC1, GG1, and TT1 6-mer subsets, the dispersion values cannot significantly distinguish the differences among three acceptor splicing modes (Figure 6C). These results indicate that the sequence composition characterized by the dispersion feature in the downstream region showed significant differences among the three splicing modes.

### 3.3. Differences Analysis of Three Alternative Acceptor Splicing Sub-Modes

The alternative acceptor splicing sequence set was classified into normal, exonic, and intronic sub-modes. The sequence numbers of the three sub-modes are 12,356, 50,497 and 144,098, respectively. The dispersion values of 16 XY1 6-mer subsets were calculated at each site and the distributions of the three sub-modes are shown in Figure 7.

Generally, the dispersion distributions of XY1 6-mers are similar between intronic and normal sub-modes, while both show obvious differences compared to the exonic sub-mode (Figure 7). To explore the sequence composition differences among the three alternative splicing modes, we performed the paired sample *t*-tests on the dispersion values of XY1 subsets at each site in the upstream, core, and downstream regions to reveal the differences among the three alternative splicing sub-modes.

#### 3.3.1. Upstream Region of Three Alternative Acceptor Splicing Sub-Modes

We extracted the dispersion values of XY1 6-mer subsets in 44 sites from sites −50 to −7 to perform the paired sample *t*-test among the three kinds of splicing sub-modes. The degree of freedom for the paired sample *t*-test is 43. The average dispersion values of XY1 6-mers in the upstream region and the results of the significant test are shown in Figure 8.

In the upstream region, the paired sample *t*-tests showed that the dispersion differences in eleven XY1 6-mer subsets are significant among the three alternative acceptor splicing sub-modes. The eleven subsets are AC1, CA1, CG1, GC1, CT1, TC1, AG1, CC1, GG1, GT1, and TT1 6-mers (Figure 8A). For GA1 and TG1 6-mer subsets, the paired sample *t*-tests showed that the dispersion differences significantly distinguish the exonic sub-mode from the other two sub-modes. For TA1 6-mer subsets, the paired sample *t*-tests showed that the dispersion differences significantly distinguish the normal sub-mode from the other two sub-modes (Figure 8B). For AA1 and AT1 6-mer subsets, the dispersion values cannot significantly distinguish the differences among three alternative acceptor splicing sub-modes (Figure 8C). These results indicate that the sequence composition characterized by the dispersion feature in the upstream region also showed significant differences among the three alternative splicing sub-modes.

#### 3.3.2. Core Region of Three Alternative Acceptor Splicing Sub-Modes

We extracted the dispersion values of XY1 6-mer subsets in 7 sites from sites −6 to +1 to perform the paired sample *t*-test among the three kinds of splicing sub-modes. The degree of freedom for the paired sample *t*-test is 6. The average dispersion values of XY1 6-mers in the core region and the results of the significant test are shown in Figure 9.

In the core region, the paired sample *t*-tests showed that the dispersion differences in the CC1 6-mer subset are significant among the three alternative acceptor splicing sub-modes (Figure 9A). For AA1, GC1, AT1, GT1, and TA1 6-mer subsets, the paired sample *t*-tests showed that the dispersion differences significantly distinguish the exonic sub-mode from the other two sub-modes (Figure 9B). For the other ten 6-mer subsets, the dispersion values cannot significantly distinguish the differences among three alternative acceptor splicing sub-modes. The ten subsets are AC1, TT1, CA1, CG1, CT1, GA1, GG1, TC1, TG1 and AG1 6-mers (Figure 9C). These results indicated that the sequence composition in the core region is more conserved among the three alternative splicing sub-modes. However, the sequence composition of the exonic sub-mode in the core region showed obvious differences compared to the other two sub-modes.

#### 3.3.3. Downstream Region of Three Alternative Acceptor Splicing Sub-Modes

We extracted the dispersion values of XY1 6-mer subsets in 44 sites from sites +2 to +45 to perform the paired sample *t*-test among the three kinds of splicing sub-modes. The degree of freedom for the paired sample *t*-test is 43. The average dispersion values of XY1 6-mers in the downstream region and the results of the significant test are shown in Figure 10.

In the downstream region, the paired sample *t*-tests showed that the dispersion differences in AA1, AT1, TA1, CT1, and TG1 6-mer subsets are significant among the three alternative acceptor splicing sub-modes (Figure 10A). For AG1, GA1, GT1, and TC1 6-mer subsets, the paired sample *t*-tests showed that the dispersion differences significantly distinguish the exonic sub-mode from the other two sub-modes. For CA1 and CC1 6-mer subsets, the paired sample *t*-tests showed that the dispersion differences significantly distinguish the intronic sub-mode from the other two sub-modes. For CG1 6-mer subset, the paired sample *t*-tests showed that the dispersion differences significantly distinguish the normal sub-mode from the other two sub-modes (Figure 10B). For AC1, GC1, GG1, and TT1 6-mer subsets, the dispersion values cannot significantly distinguish the differences among three alternative acceptor splicing sub-modes (Figure 10C). These results indicated that the sequence composition characterized by the dispersion feature in the downstream region also showed significant differences among the three alternative splicing sub-modes.

## 4. Discussion

Understanding the sequence composition information of donor and acceptor splicing regions is critical for unraveling the mechanisms underlying splicing regulation. The sequence composition information includes two important aspects: base bias and base correlation. The information of long-range base correlations (typically spanning 4–8 bp) is very important in functional sequences. The 6-mer analysis can capture the information of long-range base correlation. However, long k-mer analysis, such as 6-mer with a number of 4^6^ = 4096, may pose serious challenges due to the large number of frequency values, making it difficult to derive clear and simple conclusions. To reduce the analysis complexity, we proposed a classification method to divide all 6-mers into 32 XY 6-mer subsets. Finally, we selected the features of 16 XY1 6-mer subsets for the 6-mer analysis. We used the dispersion feature of XY1 6-mer subsets to analyze the sequence composition and obtained effective signal patterns in acceptor splicing regions.

In addition, we observed significant differences between the alternative mode and both the constitutive and common modes. These differences may stem from the distinct locations of the splice sites: constitutive and common sites are located at exon–intron junctions, whereas alternative splice sites can occur within introns or exons, leading to differences in the surrounding background sequences.

In our previous study, we used the separability features of 16 XY1 6-mer subsets to analyze the sequence composition of the acceptor splicing regions and obtained meaningful conclusions. The separability features are more focused on the information of base bias. Here, the dispersion features are more focused on the information of base correlation. We found that both separability and dispersion features are more effective in revealing the sequence composition of the acceptor splicing region. We are convinced that the two kinds of feature sets will enhance the recognition and prediction accuracy of different acceptor splicing modes when they are added to the known dataset in machine learning models.

Combining our previous and current studies, we consider that dispersion features provide complementary insights to those obtained from separability features, highlighting their potential to uncover additional sequence-specific patterns. Next, we will extend the use of these two kinds of features to investigate the donor splicing regions in human genes. We expect to reveal new sequence composition rules and further validate the regulatory relationships between donor and acceptor splicing sites in human genes. However, our study did not involve non-canonical splicing patterns because there is no available database that systematically classifies the splicing modes in both donor and acceptor regions. This will be studied in the future as more data become available. For instance, we plan to construct datasets that include non-canonical splicing sites, allowing us to better understand the diversity of splicing mechanisms.

In addition to expanding our dataset, we aim to apply the 6-mer analysis method to other species, such as mouse, to explore the conserved and evolutionary features of splicing sequences. This comparative analysis will help identify species-specific splicing modes and the conserved mechanisms shared across species. Furthermore, we plan to investigate splicing-related diseases by combining the separability and dispersion features. This approach will allow us to identify sequence-based biomarkers associated with specific splicing abnormalities and provide insights into their roles in human diseases. These studies will lay the groundwork for a better understanding of splicing mechanisms and their implications for health and disease.

## 5. Conclusions

We extracted a dataset of the acceptor splicing sequences for the canonical splicing site (GT-AG) from the multiple transcripts in human genes. According to the known definition of splicing events [4,5,6], we classified the acceptor splicing sequences into common, constitutive, and alternative modes, and further classified the alternative mode into normal, exonic, and intronic sub-modes. To sufficiently characterize the information of base correlation, we used the dispersion feature of 16 kinds of XY1 6-mer subsets to explore the sequence composition and the composition differences in total acceptor splicing sequences and three splicing modes as well as three alternative splicing sub-modes.

We observed that the dispersion feature of 16 XY1 6-mer subsets can characterize the sequence composition differences in the upstream, core, and downstream regions of acceptor splicing. The 6-mer composition of 12 XY1 subsets in the upstream region, 6 XY1 subsets in the core region, and 14 XY1 subsets in the downstream region showed strong conservation or strong base correlation. We found that the dispersion feature of XY1 6-mer subsets can reveal differences among the three acceptor splicing modes. There are eleven XY1 6-mers in the upstream region, two XY1 6-mers in the core region, and nine XY1 6-mers in the downstream region that can significantly distinguish the three splicing modes. Furthermore, the dispersion feature of XY1 6-mer subsets can also reveal the differences among the three alternative acceptor splicing sub-modes. There are eleven XY1 6-mers in the upstream region, one XY1 6-mer in the core region, and five XY1 6-mers in the downstream region that can significantly distinguish the three splicing modes. The exonic sub-mode showed obvious differences compared with the other two sub-modes in the whole acceptor splicing region.

## Figures and Tables

**Figure 1 biology-14-00206-f001:**
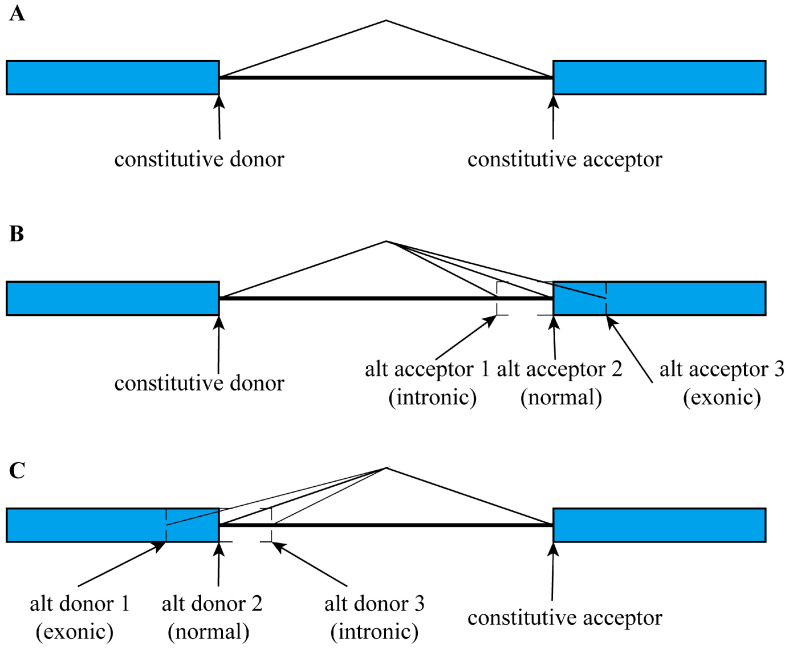
Classification scheme of the three acceptor splicing modes. (**A**) Common acceptor splicing mode. (**B**) Alternative acceptor splicing mode. (**C**) Constitutive acceptor splicing mode.

**Figure 2 biology-14-00206-f002:**
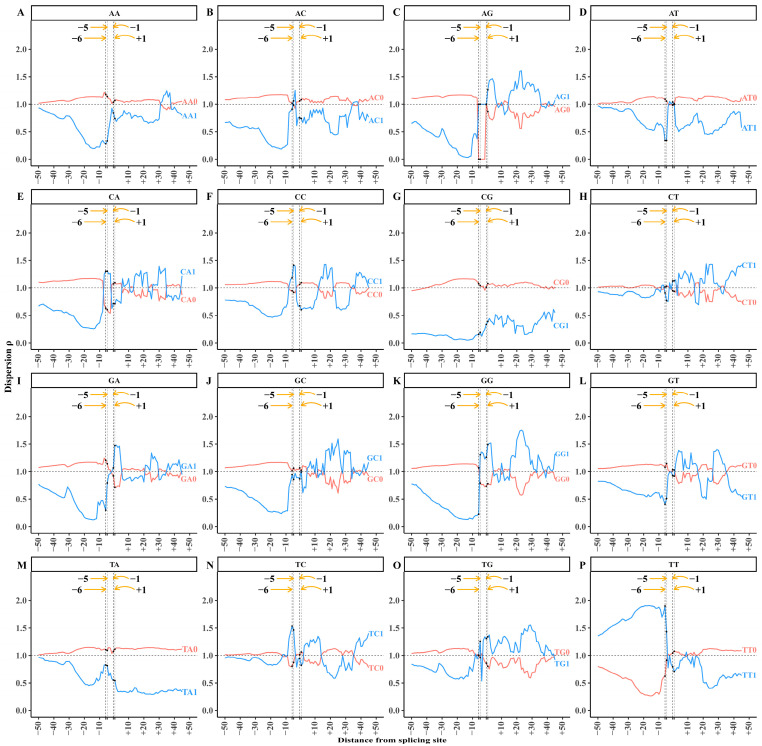
Dispersion distributions of XY1 and XY0 6-mer subsets at each site in the acceptor splicing sequences. (**A**) AA0 and AA1 6-mer subsets. (**B**) AC0 and AC1 6-mer subsets. (**C**) AG0 and AG1 6-mer subsets. (**D**) AT0 and AT1 6-mer subsets. (**E**) CA0 and CA1 6-mer subsets. (**F**) CC0 and CC1 6-mer subsets. (**G**) CG0 and CG1 6-mer subsets. (**H**) CT0 and CT1 6-mer subsets. (**I**) GA0 and GA1 6-mer subsets. (**J**) GC0 and GC1 6-mer subsets. (**K**) GG0 and GG1 6-mer subsets. (**L**) GT0 and GT1 6-mer subsets. (**M**) TA0 and TA1 6-mer subsets. (**N**) TC0 and TC1 6-mer subsets. (**O**) TG0 and TG1 6-mer subsets. (**P**) TT0 and TT1 6-mer subsets. The *X*-axis represents the base site of the acceptor splicing sequence. The splicing site is located between sites −1 and +1. The *Y*-axis represents the dispersion values of the XY0 and XY1 subsets at each site. Blue curves represent the XY1 6-mer subsets, and red curves represent the XY0 6-mer subsets.

**Figure 3 biology-14-00206-f003:**
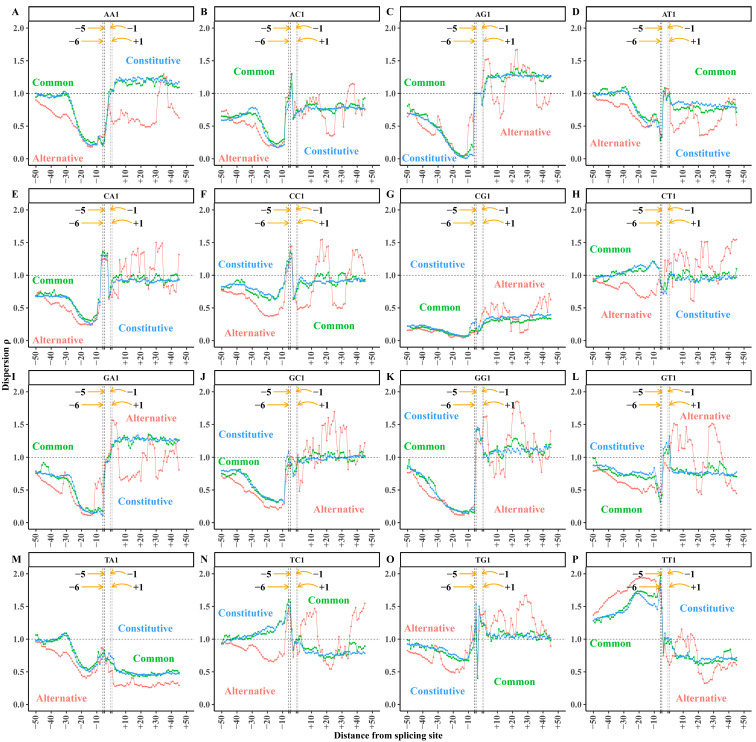
Dispersion distributions of XY1 6-mer subsets in the acceptor splicing sequences. (**A**) AA1 6-mer subset. (**B**) AC1 6-mer subset. (**C**) AG1 6-mer subset. (**D**) AT1 6-mer subset. (**E**) CA1 6-mer subset. (**F**) CC1 6-mer subset. (**G**) CG1 6-mer subset. (**H**) CT1 6-mer subset. (**I**) GA1 6-mer subset. (**J**) GC1 6-mer subset. (**K**) GG1 6-mer subset. (**L**) GT1 6-mer subset. (**M**) TA1 6-mer subset. (**N**) TC1 6-mer subset. (**O**) TG1 6-mer subset. (**P**) TT1 6-mer subset. The *X*-axis represents the base site of the acceptor splicing sequence. The acceptor splicing site is between site −1 and site +1. The *Y*-axis represents the dispersion values of XY1 6-mer subsets at each site. Green curves represent the common mode, blue curves represent the constitutive mode, and red curves represent the alternative mode.

**Figure 4 biology-14-00206-f004:**
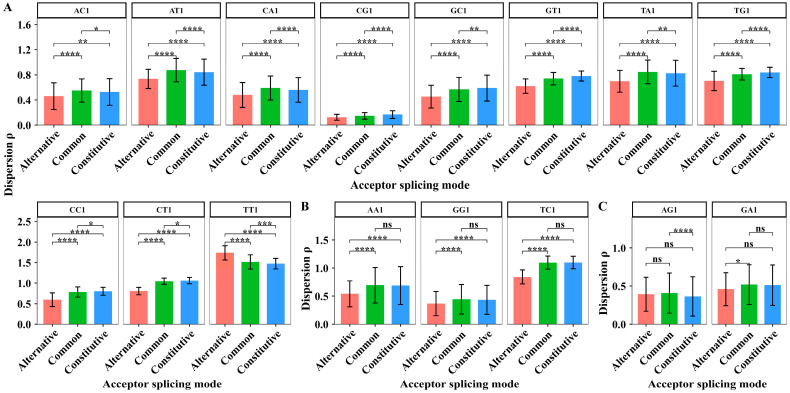
The average dispersion values and the difference analysis of the dispersion values for 16 kinds of XY1 6-mer subsets in the upstream region from sites −50 to −7 among the three kinds of acceptor splicing modes. (**A**) The XY1 6-mers with significant differences among the three modes. (**B**) The XY1 6-mers with significant differences only between two modes. (**C**) The XY1 6-mers with no significant differences among the three modes. Here, ns represents *p*-value > 0.05, * represents *p*-value ≤ 0.05, ** represents *p*-value ≤ 0.01, *** represents *p*-value ≤ 0.001, and **** represents *p*-value ≤ 0.0001.

**Figure 5 biology-14-00206-f005:**
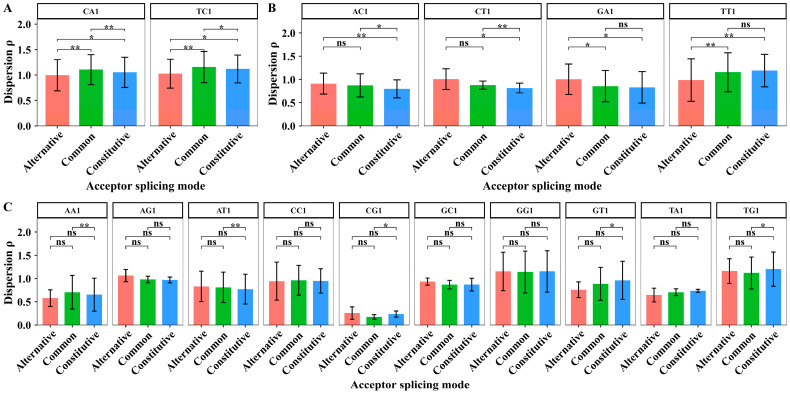
The average dispersion values and the difference analysis of dispersion values for 16 kinds of XY1 6-mer subsets in the core region from sites −6 to +1 among the three kinds of acceptor splicing modes. (**A**) The XY1 6-mers with significant differences among the three modes. (**B**) The XY1 6-mers with significant differences only between two modes. (**C**) The XY1 6-mers with no significant differences among the three modes. Here, ns represents *p*-value > 0.05, * represents *p*-value ≤ 0.05, and **. represents *p*-value ≤ 0.01.

**Figure 6 biology-14-00206-f006:**
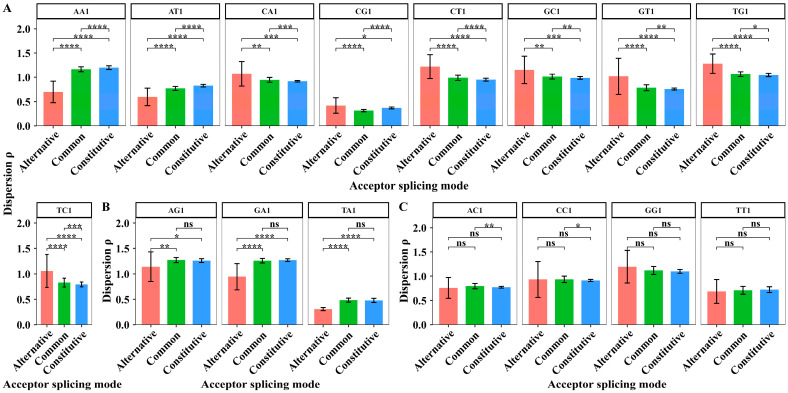
The average dispersion values and the difference analysis of dispersion values for 16 kinds of XY1 6-mer subsets in the downstream region from sites +2 to +45 among the three kinds of acceptor splicing modes. (**A**) The XY1 6-mers with significant differences among the three modes. (**B**) The XY1 6-mers with significant differences only between two modes. (**C**) The XY1 6-mers with no significant differences among the three modes. Here, ns represents *p*-value > 0.05, * represents *p*-value ≤ 0.05, ** represents *p*-value ≤ 0.01, *** represents *p*-value ≤ 0.001, and **** represents *p*-value ≤ 0.0001.

**Figure 7 biology-14-00206-f007:**
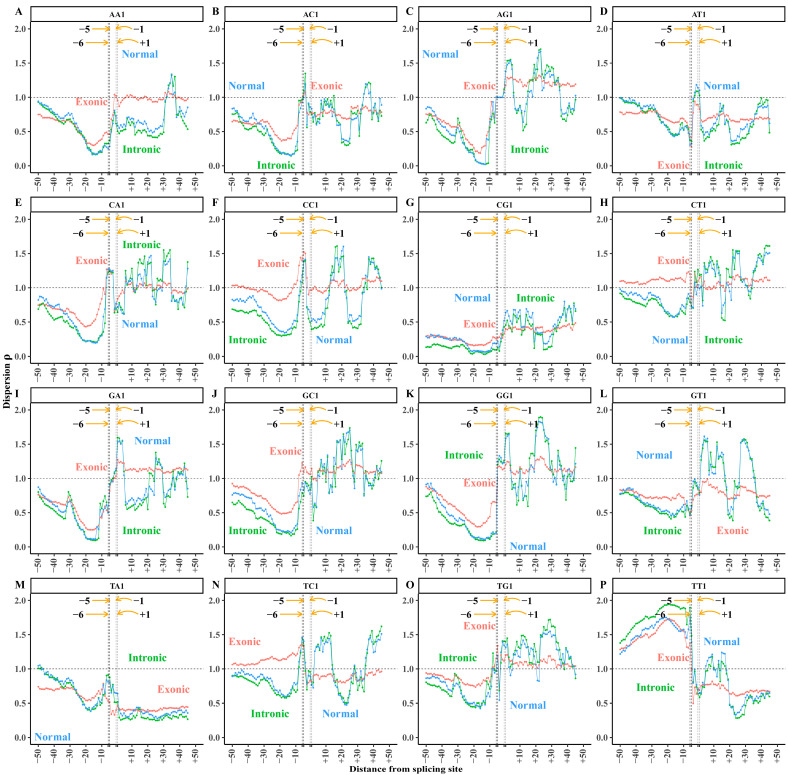
Dispersion distributions of XY1 6-mer subsets among the normal, exonic, and intronic acceptor splicing sequences. (**A**) AA1 6-mer subset. (**B**) AC1 6-mer subset. (**C**) AG1 6-mer subset. (**D**) AT1 6-mer subset. (**E**) CA1 6-mer subset. (**F**) CC1 6-mer subset. (**G**) CG1 6-mer subset. (**H**) CT1 6-mer subset. (**I**) GA1 6-mer subset. (**J**) GC1 6-mer subset. (**K**) GG1 6-mer subset. (**L**) GT1 6-mer subset. (**M**) TA1 6-mer subset. (**N**) TC1 6-mer subset. (**O**) TG1 6-mer subset. (**P**) TT1 6-mer subset. The *X*-axis represents the base site of the acceptor splicing sequence. The acceptor splicing site is between site −1 and site +1. The *Y*-axis represents the dispersion values of XY1 6-mer subsets corresponding to each site. Green curves represent the intronic mode, blue curves represent the normal mode, and red curves represent the exonic mode.

**Figure 8 biology-14-00206-f008:**
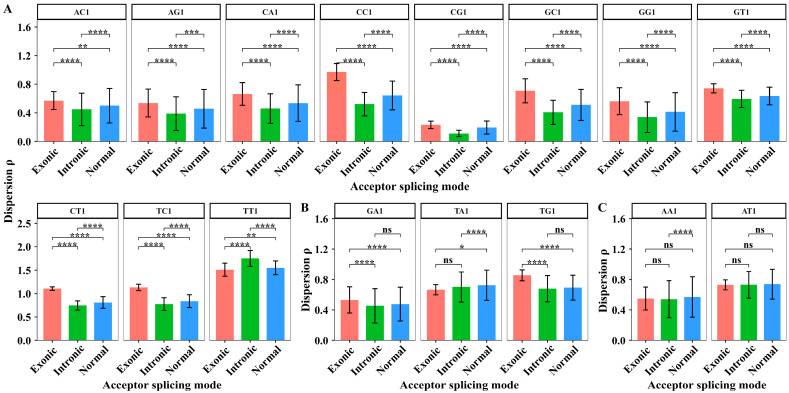
The average dispersion values and the difference analysis of dispersion values for 16 kinds of XY1 6-mer subsets in the upstream region among the three kinds of alternative acceptor splicing sub-modes. (**A**) The XY1 6-mers with significant differences among the three sub-modes. (**B**) The XY1 6-mers with significant differences only between two sub-modes. (**C**) The XY1 6-mers with no significant differences among the three sub-modes. Here, ns represents *p*-value > 0.05, * represents *p*-value ≤ 0.05, ** represents *p*-value ≤ 0.01, *** represents *p*-value ≤ 0.001, and **** represents *p*-value ≤ 0.0001.

**Figure 9 biology-14-00206-f009:**
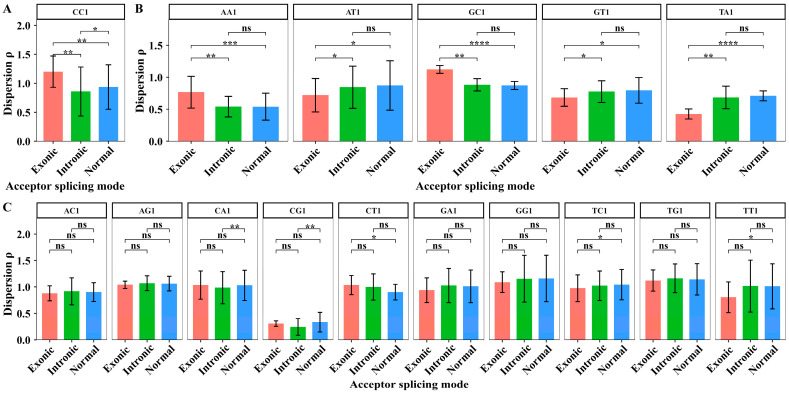
The average dispersion values and the difference analysis of dispersion values for 16 kinds of XY1 6-mer subsets in the core region among the three kinds of alternative acceptor splicing sub-modes. (**A**) The XY1 6-mers with significant differences among the three sub-modes. (**B**) The XY1 6-mers with significant differences only between two sub-modes. (**C**) The XY1 6-mers with no significant differences among the three sub-modes. Here, ns represents *p*-value > 0.05, * represents *p*-value ≤ 0.05, ** represents *p*-value ≤ 0.01, *** represents *p*-value ≤ 0.001, and **** represents *p*-value ≤ 0.0001.

**Figure 10 biology-14-00206-f010:**
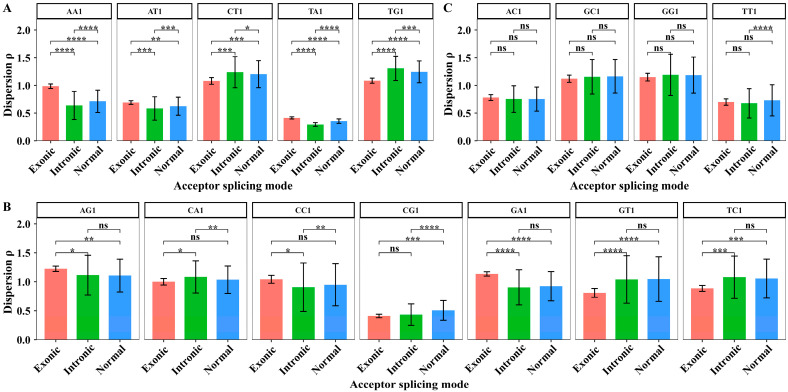
The average dispersion values and the difference analysis of dispersion values for 16 kinds of XY1 6-mer subsets in the downstream region among the three kinds of alternative acceptor splicing sub-modes. (**A**) The XY1 6-mers with significant differences among the three sub-modes. (**B**) The XY1 6-mers with significant differences only between two sub-modes. (**C**) The XY1 6-mers with no significant differences among the three sub-modes. Here, ns represents *p*-value > 0.05, * represents *p*-value ≤ 0.05, ** represents *p*-value ≤ 0.01, *** represents *p*-value ≤ 0.001, and **** represents *p*-value ≤ 0.0001.

**Table 1 biology-14-00206-t001:** Dataset of acceptor splicing sequences in human genes.

Acceptor Splicing Mode	Alternative Splicing Sub-Mode	Number of Sequences
Common splicing		57,938
Constitutive splicing		48,414
Alternative splicing		206,951
Alternative splicing	Normal splicing	12,356
	Exonic splicing	50,497
	Intronic splicing	144,098

## Data Availability

All scripts used in this research are publicly available in the GitHub repository at https://github.com/SiYangming/Donor_Acceptor_XY_6mer (accessed on 31 December 2024) for reproducibility. All acceptor splicing sequences referenced in this study are accessible at https://doi.org/10.6084/m9.figshare.26892364.v2 (accessed on 2 September 2024).

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
