# Peer review of "Difference Analysis Among Six Kinds of Acceptor Splicing Sequences by the Dispersion Features of 6-mer Subsets in Human Genes"

_biology, 2025, doi:10.3390/biology14020206_

Round 1
Reviewer 1 Report
Comments and Suggestions for Authors
1) The introduction should be shortened
2) 3.1.2. Upstream Region. You should avoid citing your own results
3) I would recommend that the authors label the panels with numbers in the figures and refer to the text not just to the figure, but more specifically.
4) The results are described in too much detail and should be shortened. Whereas the discussion of the results is too short and is not really a discussion, since it does not contain references to other works.
5) The conclusion is too long. Some of the information could have been used in the discussion of the results.
Reviewer 2 Report
Comments and Suggestions for Authors
-The study focuses predominantly on canonical GT-AG splicing sites, potentially overlooking non-canonical patterns, which could limit the generalizability of the findings.
-Incorporate non-canonical splicing sites to broaden the scope and applicability of findings.
-Although alternative, constitutive, and common modes are distinguished, the biological implications of observed differences remain underexplored.
-Provide a deeper discussion on how dispersion patterns relate to functional splicing events and their potential implications in gene expression regulation or diseases.
-A comparative analysis with other organisms, such as mice or plants, could validate findings and reveal conserved splicing mechanisms.
-Incorporate machine learning models, such as neural networks, to validate the predictive power of dispersion features for splicing site classification.
Reviewer 3 Report
Comments and Suggestions for Authors
The manuscript is appealing and it would gain attention of readers or scientists related to bioinformatics and biotechnology. However, few comments need to be addressed before publication
Please provide the citation in the methodology section
what is the biological importance of 16 XY1 6-mers region?
What were the dispersion metrics for the study?
How do compositional differences in receptor splicing mode help biotechnologists for gene therapy?
Which 6-mers region is rich in XY1 or presents important sequence of regulatory regions?
Round 2
Reviewer 1 Report
Comments and Suggestions for Authors
All my comments have been taken into account. The quality of the article has been significantly improved. The article can be accepted in its current form.